# Involvement of Target of Rapamycin (TOR) Signaling in the Regulation of Crosstalk between Ribosomal Protein Small Subunit 6 Kinase-1 (RPS6K-1) and Ribosomal Proteins

**DOI:** 10.3390/plants12010176

**Published:** 2023-01-01

**Authors:** Achala Bakshi, Mazahar Moin, Meher B. Gayatri, Aramati B. M. Reddy, Raju Datla, Maganti S. Madhav, Pulugurtha B. Kirti

**Affiliations:** 1Indian Institute of Rice Research, Rajendranagar, Hyderabad 500030, Telangana, India; 2Global Institute for Food Security, Saskatoon, SK S7N 0W9, Canada; 3Agri Biotech Foundation, PJTS Agricultural University Campus, Rajendranagar, Hyderabad 500030, Telangana, India; 4Department of Animal Biology, University of Hyderabad, Hyderabad 500046, Telangana, India; 5Central Tobacco Research Institute, Rajahmundry 533105, Andhra Pradesh, India; 6Department of Plant Sciences, University of Hyderabad, Hyderabad 500046, Telangana, India

**Keywords:** target of rapamycin, ribosomal proteins large subunit genes, ribosomal proteins small subunit genes

## Abstract

The target of rapamycin (TOR) protein phosphorylates its downstream effector p70kDa ribosomal protein S6 kinases (S6K1) for ribosome biogenesis and translation initiation in eukaryotes. However, the molecular mechanism of TOR-S6K1-ribosomal protein (RP) signaling is not well understood in plants. In the present study, we report the transcriptional upregulation of ribosomal protein large and small subunit (*RPL* and *RPS*) genes in the previously established *TOR* overexpressing transgenic lines of rice (in *Oryza sativa* ssp. *indica*, variety BPT-5204, TR-2.24 and TR-15.1) and of *Arabidopsis thaliana* (in Col 0 ecotype, ATR-1.4.27 and ATR-3.7.32). The mRNA levels of *RP* genes from this study were compared with those previously available in transcriptomic datasets on the expression of *RP*s in relation to TOR inhibitor and in the *TOR*-RNAi lines of *Arabidopsis thaliana*. We further analyzed TOR activity, i.e., S6K1 phosphorylation in SALK lines of Arabidopsis with mutation in *rpl6*, *rpl18*, *rpl23*, *rpl24* and *rps28C,* where the *rpl18* mutant showed inactivation of S6K1 phosphorylation. We also predicted similar putative Ser/Thr phosphorylation sites for ribosomal S6 kinases (RSKs) in the RPs of *Oryza sativa* ssp. *indica* and *Arabidopsis thaliana.* The findings of this study indicate that the TOR pathway is possibly interlinked in a cyclic manner via the phosphorylation of S6K1 as a modulatory step for the regulation of RP function to switch ‘on’/‘off’ the translational regulation for balanced plant growth.

## 1. Introduction

The TOR is a central regulator of ribosome biogenesis and protein synthesis in eukaryotes [1,2]. Several TOR substrates involved in the translational regulation have been identified previously, which include S6 kinases (S6Ks), eIF4E-binding proteins (4E-BPs), eIF4G initiation factors and La-related protein 1 (LARP1) [3,4,5,6,7]. S6Ks are members of the AGC family (which includes PKA, PKG and PKC) and are conserved substrates of TOR signaling [8,9]. Both animals and plants have two members of S6Ks, which are S6K1 and S6K2 [9,10]. TOR phosphorylates p70kDa ribosomal S6 kinase (S6K)-1 in Thr-389 residue and Thr-449 residue in animals and plants, respectively [11,12,13,14,15,16]. The activation of mammalian S6K requires phosphorylation at three conserved sites, namely the T-loop, the TM site and the HM site, which show high homology to plant S6Ks, except that the four proline residues directing Ser/Thr sites in the C-terminal domain are absent in plants. In contrast, plant S6Ks can be activated by phosphorylation at two of these three sites [10].

Multisite phosphorylation fully activates S6K1, which further phosphorylates several targets including ribosomal protein S6 in 40S ribosome. After the assembly of 40S with 60 S converts into an 80S ribosome, S6K is dephosphorylated and dissociated from the complex allowing the ribosome to proceed for the peptide elongation step [17]. Global phosphoproteomic study has predicted that LARP1 plays an important intermediate role in the translational regulation of the 5′-terminal oligopyrimidine tract (5′TOP) mRNA, when TOR is inactivated and the S6K-pT449/S6K ratios are noticeably lower [6,7]. Ribosomal proteins (RPs) are heterogeneous in nature and each RP is encoded by one of the multiple homologous partners for a given protein [18], where the function of which is determined and influenced by the tissue and the environmental conditions of the plant. The 80 S ribosomal large and small subunits consist of a heterogeneous incorporation of RPs and various rRNAs. Some cytosolic *RP* genes are also reported as 5′TOP mRNAs, which are regulated by either TOR-LARP1 or TOR-S6K1 signaling [7]. However, the molecular mechanisms of other S6K1 substrates are unknown and the interaction between RPs with S6K1 and their contribution to translation initiation is poorly understood in plants. The *TOR* also plays a crucial role in the transcription of nuclear-encoded mRNAs coding for plastidic RPs in Arabidopsis [19]. The inactivation of the TOR in two *TOR*-RNAi lines of Arabidopsis resulted in a coordinated decrease in transcription and translation of plastidic ribosomal protein genes, whereas the genes coding for cytosolic ribosomal proteins were interestingly upregulated [19]. In *Saccharomyces cerevisiae*, the mechanism of *TOR*-mediated regulated *RP* genes is well studied. The TOR controls the Forkhead transcription factor (FHL1) along with its co-activator, IFH1, and a repressor, CRF1, and triggers the transcription of *RP* genes under nutrient availability and nutrient-deprived conditions in yeast [20,21]. Despite having fundamental importance, the functions of TOR signaling in the regulation of genes encoding large and small subunit RPs are not well understood in plants. We, therefore, have explored the effects of the overexpression of *TOR* in the homologous and heterologous systems in Arabidopsis and rice, respectively, on the changes in the expression levels of *RP* genes. 

We have also previously reported that rice and Arabidopsis plants ectopically overexpressing *Arabidopsis thaliana TOR* (*AtTOR*) exhibited enhanced growth, increased seed yield, leaf size and improved stress tolerance [2,22,23,24,25]. In the present study, we have therefore used these *TOR* overexpressing lines of *Oryza sativa* ssp. *indica var.* BPT-5204 and *Arabidopsis thaliana* for analyzing the mRNA levels of ribosomal proteins encoding genes. The lines represented with TR are *TOR* overexpressing lines of rice and the lines represented with ATR are *TOR* overexpressing lines of *Arabidopsis thaliana* [23,25]. In addition to the transcriptional regulation of *RP* genes, the results from a Western blot analysis in this study also show that TOR-mediated S6K1 phosphorylation is affected by a loss of RP function in Arabidopsis.

## 2. Results

### 2.1. Expression Analysis of RP Genes in the TOR-OE Lines of Rice and Arabidopsis

Although the involvement of TOR signaling in the modulation of *RP* genes transcription has been demonstrated previously in Arabidopsis [2,19,26,27], no such reports are available with regard to *TOR*-overexpressing crop-plants. We, therefore, performed an expression analysis of *RPL* and *RPS* genes in two high *AtTOR*-expressing rice lines, TR-2.24 and TR-15.1, and two Arabidopsis lines, ATR-1.4.27 and ATR-3.7.32 [23,25], to assess the effects of ectopic expression of *AtTOR* on the regulation of *RP* transcription in both of the species. In the present study, the paralogs of *RPL* and *RPS* genes were selected based on their expression levels in various developmental stages of shoots and roots of rice and Arabidopsis plants. The transcript levels of the *RP* genes were significantly upregulated in the rice transgenic lines. The genes *RPL4*, *RPL14*, *RPL18A*, *RPL19.3*, *RPL36.2*, *RPL51*, *RPS3A*, *RPS6*, *RPS6A*, *RPS25A* and *RPS30* display upregulation in the rice transgenic plants that expressed *AtTOR* ectopically (Figure 1a and Figure 2a,b). The transcript levels of *RPL18A*, *RPL19.3*, *RPL51*, *RPS25A* and *RPS30* were significantly increased up to ten-fold and *RPL4*, *RPL14*, *RPL24B*, *RPL26.1*, *RPL30e*, *RPL38A*, *RPL44*, *RPS3A*, *RPS6*, *RPS6A*, *RPS27* and *RPS27a* were upregulated by more than five-fold (Figure 1a and Figure 2a,b). The transcript levels of *RPL* and *RPS* genes were significantly upregulated in the Arabidopsis *TOR*-OE lines, ATR-1.4.27 and ATR-3.7.32, except for the genes *RPS2, RPS4, RPL12, RPL17, RPL26* and *RPL39* that exhibited lower expression levels (Figure 1b and Figure 2c,d). 

The data of RP gene expression of *AtTOR*-OE rice and Arabidopsis lines obtained using RT-qPCR analysis were compared with the reports on the transcriptome analysis of RP genes by Dong et al. (2015) and Dobrenel et al. (2016) in *TOR*-RNAi lines of Arabidopsis (Figure 3a,b and Figure 4a,b) [19,27]. The results showed that the transcript levels of the cytoplasmic genes *RPS5, RPS 8, RPS 14, RPS 20, RPS 23, RPS 29, RPS 30, RPL7, RPL13, RPL13a, RPL17, RPL26* and *RPL29* were either slightly modulated or remained unchanged. Altogether, these data strongly suggest a positive correlation between the *TOR* expression levels and RP gene transcription in plants.

Among the analyzed *RP* genes in rice (71 *RP* genes) and in Arabidopsis (73 *RP* genes), *AtTOR*-OE lines TR-2.24, TR-15.1, ATR-1.4.27 and ATR-3.7.32, the transcript levels of the 28 overlapping *RP* genes were significantly upregulated by more than 2-fold (Figure 4c). The transcript levels of *RP* genes were significantly reduced in the AZD-8055-treated Arabidopsis lines [27], suggesting a positive influence of *TOR* expression on *RP* gene expression. However, contrary to this expectation, the *TOR*-RNAi lines of Arabidopsis showed significant upregulation of *RP* gene expression [19]. AZD-8055, an ATP competitive TOR inhibitor, was screened as the strongest active site TOR inhibitor (asTORi) in Arabidopsis compared with the other second-generation TOR inhibitors TORIN1 and KU63794 (KU) [27]. These inhibitors potentially target the ATP-binding pocket of the TOR kinase domain [28]. The differences between these two studies might be caused by the mode of silencing efficiencies between the *TOR*-RNAi lines and AZD-8055 treatment, or by selection of different growth conditions used in the study. Altogether, the results indicate that the *TOR* dynamically regulates the *RP* genes transcription and plays a crucial role in ribosome biogenesis.

### 2.2. Identification of Putative Phosphorylation Sites and Protein Kinase Binding Motifs in RPL and RPS Proteins

The ribosomal S6 protein kinases’ (RSKs) family proteins are serine/threonine kinases that regulate cell growth and proliferation. The two subfamilies of RSKs, p90 RSK and p70 S6K, phosphorylate RPS6 in Ser/Thr residues for the modulation of ribosome biogenesis and protein translation [7,12,28,29]. The RPS6 phosphorylation sites via S6K are identified in Ser237, Ser 238, Ser240 and Ser241 in plants [29,30]. Previously, the RPS6 was identified as the only phosphorylated protein among the ribosome small subunit proteins but the study on post-translational modifications in the ribosomal proteins of Arabidopsis identified the phosphorylation of the RPL13 protein in Ser137 residue [31]. The TOR inactivation in the RNAi lines of Arabidopsis showed inhibition of phosphorylation of RPS6A and RPS6B proteins in Ser240 and Ser237 residue at the C-terminal of peptide sequence pSRLpSSAPAKPVAA [19]. We observed a similar peptide sequence pSKLpSSAAKA at the C-terminal of rice RPS6A and RPS6B proteins in Ser 240 and Ser 241 residues, indicating the conserved nature of phosphorylation sites in plants. Similarly, the pSKLpSQGIK peptide was predicted in the RPL6 protein in Ser5 residue. Although validation through biochemical assays is further required for their phosphorylation, the predicted peptide sequences indicated the possibility of their phosphorylation by S6K (Table 1; Appendix A). The predicted peptide sequences with the Ser/Thr phosphorylation sites were then compared with the conserved RPS6 peptide sequences in Arabidopsis. The sequence alignment of rice (*Oryza sativa* ssp. *japonica*) RPS6A/B proteins from ssp. *japonica* shared the highest similarity with *Arabidopsis thaliana* RPS6A/B proteins in their Ser/Thr phosphorylation sites in similar amino acid positions (Appendix A). The OsRPL6 protein has twenty-five Ser/Thr phosphorylation sites for various AGC kinases (PKA, PKB, PKC, PKG and RSK) when compared with the AtRPL6A/B proteins having twenty phosphorylation sites, and it exhibited maximum variation in its phosphorylation sites and its amino acid sequence was adjacent to the Ser/Thr sites. A conserved Guanine residue is present in the peptide sequence of AtRPS6 in Thr 81 residue (RGTP) and in Thr 93(PGTV) of the AtRPL6 protein. Similarly, a conserved Leucine was also observed in Ser 119 (QLSL) of AtRPL6 and Ser 109 (DLSV) of AtRPS6. The presence of conserved amino acids adjacent to the phosphorylation sites in Ser/Thr residues in AtRPS6 and other RPs indicates their interaction with RSK (S6K1/2) proteins. The other ribosomal proteins consisted of similar conserved sites such as AtRPS6-Thr 91(RTGE)/AtRPL18-Thr 72(MTGK) and AtRPS6-Thr 129 (DTEK)/AtRPL18- Thr 105(FTER). The results of sequence alignment of RPs with AtRPS6 also showed the replacement of Ser or Thr to Thr or Ser phosphorylation sites with similar peptide sequences (AtRPS6-Ser105-VSPDL to AtRPL18-Thr86-ITDDL). 

### 2.3. Genotyping of Arabidopsis Mutant Lines

T-DNA insertions in the Arabidopsis mutants were detected by performing PCRs using the left border-specific primer (LB) and the corresponding gene specific right primer primer to identify homozygous or hemizygous lines and using the Left Primer + Right Primer combination for gene-specific amplification in each mutant following the protocol described at http://signal.salk.edu/tdna_FAQs.html accessed on 14 March 2019 (Appendix A). 

The position of the T-DNA insertion in the *rpl23aA* mutant was observed in the third exon and the *rpl6* (*emb2394)* mutant had the T-DNA insertion in the first intron of the *RPL6* gene (At1g05190). The homozygous lines of the *rpl23aA* knockouts were shown earlier to have embryo lethality [32,33]. As previously reported, the homozygous *rpl6* (*emb2394)* mutant lines displayed yellow–green cotyledons [33]. The *tor/tor* homozygous allele showed the earliest insertion in the 47^th^ exon of the kinase domain. The homozygous *tor* knockout lines also showed embryo lethality and the embryo growth was arrested at the 16–32 cell stage as previously reported [2]. Similarly, the T-DNA insertion in the *rpl24* mutant was observed in the third intron of the *RPL24* gene, the *rpl18* mutant line had insertion in the third exon, the *rps28* had insertion in the second exon and the *s6k1* mutant line had insertion in the fourth exon of the gene (Appendix A). 

A set of LB and each gene-specific right primers were used separately to amplify the site of insertion from the T-DNA left border to the 3′ end of the gene in the homozygous mutant lines of *rpl6* (670bp), *rpl18* (880bp), *rpl23* (789bp), *rpl24* (853bp), *rps28* (894bp), *s6k1* (752bp) and *tor* (730bp). The WT Arabidopsis genome showed no amplification with LB + Right gene specific Primers, showing the specificity by depicting the absence of T-DNA. In each case, the gene-specific Left Primers + Right Primers were used to amplify the *RPL6* (1201bp), *RPL18* (1194bp), *RPL23* (1156bp), *RPL24* (1200bp), *RPS28* (1271bp), *S6K1* (1105bp) and *TOR* (1054bp) genes without T-DNA insertion in the WT, whereas the primers had no amplification in the mutant lines (Appendix A). Furthermore, the transcript levels of *S6K*1, *TOR*, *RPL6, RPL18, RPL23, RPL24* and *RPS28* genes were analyzed in the total RNA extracted from 4DAG seeds of the SALK lines. The levels of the *S6K*1, *TOR*, *RPL6, RPL18, RPL23, RPL24* and *RPS28* transcripts displayed a more strongly reduced expression than the WT (Appendix A). 

### 2.4. Ribosomal Protein Inhibition Modulates Feedback Regulation of S6K1 Phosphorylation

The S6K1 phosphorylation in Threonine 449 was strongly inhibited in the *TOR*-deficient RNAi silenced line (35-7) of Arabidopsis when compared to the WT (Col0) and was increased in the *TOR*-overexpressing line G548, suggesting that *TOR* kinase expression levels are major effectors of regulating the S6K1 phosphorylation in Arabidopsis [13,22]. To gain more insights into the involvement of RPs in the phosphorylation of the S6K protein, we performed a Western blot analysis of S6K1 in Arabidopsis insertional mutants for some of the important ribosomal proteins and observed that the mutation of *RPL* and *RPS* genes in Arabidopsis corroborated the differential regulation of S6K1 phosphorylation in Arabidopsis. Simultaneously, the mutation in *RP*s also resulted in the loss of total S6K1 stability besides its phosphorylation. The *TOR* and *S6K1* mutants were used as negative controls and both mutants showed inhibited S6K1 phosphorylation and reduced stability of total S6K1 protein. However, the *TOR* mutant showed slight phosphorylation of the S6K1 protein in comparison with the *S6K1* mutant, in which the phosphorylation was completely inhibited. The *rpl6* mutants had an equally phosphorylated S6K1 protein, whereas the phosphorylation of S6K1 in *rpl23a*, *rpl24*, *rpl24a* and *rps28a* mutants was reduced to some extent. The stability of total S6K1 protein in *rpl6*, *rpl18a* and *rpl23a* was almost similar to that in the WT sample, whereas the *rpl6*, *rpl18a* and *rpl23a* mutants had increased stability of total S6K1 protein. The mutation of *rpl18* had completely inhibited S6K1 phosphorylation and the *rpl24a* mutant had moderate inhibition, whereas the mutation in *rpl23* and *rps28* had no effect on S6K1 phosphorylation (Figure 5a–e). The WT (Col0) protein was used as a positive control for phosphorylation study and GAPDH was used as an endogenous loading control (Figure 5c). Our Western blot results clearly suggest that ribosomal proteins are interlinked and involved in the regulation of S6K1 phosphorylation in plants, as with the animal systems. 

### 2.5. Identification of Putative Networking Partners of TOR-S6K1-RP Signaling

The protein–protein interactions (PPI) network were predicted between TOR, S6K1, S6K2, RPL6, RPL18, RPL23, RPL24 and RPS28 using the STRING v11 database search tool (Appendix A). STRING v11 identifies the PPI network based on the curated literature, experimentally determined/predicted interactions, co-expression or co-occurrence of proteins and the protein homology. The PPI networks of RPL6, RPL18, RPL23, RPL24 and RPS28 were also predicted separately (Appendix A). The PPI between TOR, S6K1, S6K2, RPS6A, RPS6B, RPL6, RPL18, RPL23.1, RPL23.2, RPL24A, RPL24B and RPS28 proteins showed interactions between 115 nodes with 5707 edges with an average 99.3 node degree, average local clustering coefficient of 0.971 and <1.0 × 10^−16^ *p*-value of the PPI enrichment (Appendix A). The 50S RPL6 (emb2394, AT1G05190.1) protein showed interaction with a confidence level of ≥0.90 with the RPL19e (emb2386, AT1G02780.1) family protein, a probable ribosome biogenesis protein RLP24 (AT2G44860.1, which is involved in the biogenesis of the 60S ribosomal subunit) and *EMB2207* (AT1G43170.8) which encodes a universal cytoplasmic ribosomal protein uL3 family. RPL18 (AT1G29965) also interacts with RPL24 (AT2G36620) and RPS28C (AT5G64140). The RPL23 (AT2G39460) protein interacts with proteins essential for ribosome assembly, such as RPL24 (AT2G36620), RPS17 (AT3G10610), ATARCA (AT1G18080.1, a Transducin/WD40 repeat-like super family protein, which is also a major component of the RACK1 regulatory proteins that play a role in multiple signal transduction pathways and are involved in the MAPK cascade scaffolding in the protease IV), RACK1B_AT (AT1G48630.1, a receptor for activated C kinase 1B, whose function is to shuttle activated protein kinase C to different subcellular sites and may also function as a scaffold through physical interactions with other proteins), and the RACK1 subfamily translation elongation factor EF1B/ribosomal protein S6 family protein, which binds together with S18 to 16S ribosomal RNA (AT1G64510), and also binds to a S6K1 homolog (AT3G08850.1). Similarly, RPL24 and RPS28C (AT5G64140) interact with ribosomal constituent 40S RPSa-1, which is required for the 40S ribosome subunit assembly and processing of the 20S rRNA- precursor to mature 18S rRNA, RPuS2 family protein P40 (AT1G72370.1), LOS1 (RPS5/elongation factor G/III/V family protein, which catalyzes the GTP-dependent ribosomal translocation step during translation elongation), EIF3G1 (AT3G11400.2, an eukaryotic translation initiation factor 3 subunit G and RNA-binding component of the eukaryotic translation initiation factor 3 (eIF-3) complex, which is involved in the protein synthesis of a specialized repertoire of mRNAs and, together with other initiation factors, stimulates the binding of mRNA and methionyl-tRNAi to the 40S ribosome), ATARCA (AT1G18080.1), RPL10 (AT3G11250.1), zinc-binding ribosomal protein family protein (AT3G10950), RPS17 (AT3G10610), RPS26e (AT2G40590), RPS5 (AT2G41840), RPL16A (AT2G42740.1) and RPL18e/L15 (AT2G47570). The S6K1 and S6K2 proteins also interact with the LOS1 protein with a confidence level of 0.499. The LOS1 appears to be a downstream effector of the TOR signaling pathway via interaction with the S6K1 or S6K2 protein (Appendix A). The S6K1 also shows interaction with a chloroplastic homolog of *RPL23* (At4g18520) containing a pentatricopeptide repeat (PPR) protein. The binding of the S6K1 protein with the RPL23 having the ≥0.90 confidence level in the predicted PPI showed a possible feedback regulation of S6K1 via direct binding of the RPs. 

## 3. Discussion

Arabidopsis *RP* genes are multigene families of more than two members [34]. The expression of multiple *RP* genes of a family is required for high translational demand during plant development. We performed a comparative analysis of transcriptional regulation of *RP*s via the TOR pathway in rice and Arabidopsis. Our expression analysis of *RP* genes clearly suggested that *TOR* overexpression significantly upregulates the transcription of *RP* genes in two diverse plant species (Figure 3a,b). Dong et al. (2015) observed that 114 *RP* genes have been shown in the differentially expressed genes (DEGs) and that these genes were downregulated in the Arabidopsis seedlings treated with active site *TOR* inhibitors (as*TOR*is) and AZD-8055, except for one *RP* gene which showed upregulation [27]. The TOR inactivation in *TOR*-RNAi Arabidopsis lines resulted in a coordinated downregulation of the transcription and translation of nuclear-encoded mRNAs coding for plastidic RPs and a lower phosphorylation of the conserved Ser240 residue in the C-terminal region of the 40S ribosomal protein S6 (RPS6) [19]. These results suggest that *TOR* acts differently in the regulation of *RP* genes’ transcription when overexpressed or inhibited.

Our western blot analysis showed unchanged TOR activity in *rpl6*, *rpl23* and *rpl24* mutant lines, except for in the *rpl18* mutant line which showed inactivation of S6K1 phosphorylation in Thr-449. Previous studies in plants and animals suggest the importance of extra-ribosomal association of RPs with RNAs and other proteins and provide a link for the possible interaction with the S6K1 for translational control [17,18,27,29,35,36,37]. Bacillus subtilis, a bacterium, reportedly requires the essential binding of RPL6 with a GTPase (RbgA) for the assembly of a ribosome large subunit [38]. The studies on S6K-RP interactions have been well documented in the animal systems [38,39]. In animal cells, RPS6 is associated with mRNAs of 5’-TOP tract such as RPL11 and RPS16 and negatively regulates their translation [39]. The association of ribosomal proteins such as RPL6, RPL18 and RPL24 along with other RPs has been reported as an essential step in translational transactivation in plants and animals upon viral infection [40,41,42,43]. Although there is no supporting evidence of an interaction between the S6K1 and 60S large subunit of ribosome in plants, several reports have demonstrated that S6K1 can also interact with other RPs. A co-immunoprecipitation assay predicted RPS3, RPS6, RPS7, RPS10, RPS11 and RPS17 and RPL13A, RPL18, RPL18A, RPL19 and RPL23 as S6K-interacting proteins with conserved phosphorylation sites, RXRXXT/S [44]. Additionally, a study showed that mutation or loss of RPS19 and other RPs induces S6K phosphorylation with an increase in ROS (reactive oxygen species) levels in zebrafish [45]. Similarly, the silencing of RPS27L led to autophagy in mouse fibroblasts and human breast cancer cells via the inhibition of S6K1 phosphorylation and mTOR Complex1 activity [46]. However, these studies were conducted using animal systems and analogous data are not available in plant systems. In addition, the binding of bacterial RPL18 to L5 and 5S rRNA is an essential step for the final phase of ribosome assembly [47,48,49]. Additionally, the TOR inactivation in Arabidopsis *TOR*-RNAi showed decreased cytoplasmic *RPL18a*- transcript levels [19]. This suggests that RPL18 plays an important role in ribosome biogenesis and translation. However, S6K1 phosphorylation might also be affected by several other protein kinases (PKA or PKB), its regulation is variable in plant systems and the exact mechanism of this is still not known, but repeated events of activation of S6Ks via multisite phosphorylation and its inactivation by dephosphorylation can switch the regulation of catalytic activity on or off and can influence target substrate selection and specificity [17,50]. Taking a cue from these results, the first possibility of unchanged S6K1 phosphorylation in other *rpl* mutants could be the occurrence of the dose compensation effect via the function of other paralogs of these genes. The second possibility could be the mutation in the rpl18 protein function which might be reflected in the feedback regulation of S6K1 phosphorylation in Thr-449 residue, also suggesting a possible interaction between the other 60S large ribosomal subunit RPs and the S6K1 protein. Additionally, a third possibility of impaired ribosome biogenesis and improper association in the rpl18 mutant line might lead to the dephosphorylation of S6K1. However, further investigation about plant RPs’ structures and binding partners will uncover their association with various pathways (Figure 6).

## 4. Materials and Methods

### 4.1. Generation of TOR-OE Lines of Rice and Arabidopsis

The full-length *TOR* cDNA (7.4 kb) was amplified from *Arabidopsis thaliana* (Col 0) and cloned into pEarleyGate-203 [2]. The transgenic lines overexpressing the full-length *Arabidopsis thaliana TOR* (*AtTOR*) gene in rice (*Oryza sativa*, BPT-5204 variety of ssp. *indica*) and *Arabidopsis thaliana* (Col 0 ecotype) were developed using an in-planta transformation and floral dip method, respectively [23,25]. The high *TOR*-expressing transgenic lines of rice (TR-15.1 and TR-2.24) and Arabidopsis (ATR-1.4.27 and ATR-3.7.32) were used in the expression analysis of *RP* genes [23,25].

### 4.2. Growth Conditions of Rice and Arabidopsis Lines

The seeds of rice and Arabidopsis *TOR*-OE lines, TR-2.24, TR-15.1 and ATR-1.4.27, ATR-3.7.32, respectively, and their corresponding wild type controls (WTs) were surface sterilized with 4% sodium hypochlorite for 20 min followed by three washes with sterile double-distilled water. The rice seeds were germinated on solid Murashige and Skoog (MS) medium and were grown at 28 ± 2 °C in 16 h light/8 h dark photoperiods. The seeds of *Arabidopsis thaliana* SALK-T-DNA mutant lines, *TOR*-OE lines (ATR-1.4.27 and ATR-3.7.32) and WT plants were surface sterilized and grown on solid ½ strength of MS medium at 22 ± 2 °C of temperature and 100–150 μmol m^−2^ s^−1^ of light intensity following 16/8 h of light/dark photoperiods. Seven-day-(7 d)-old rice and Arabidopsis *TOR*-OE transgenic lines and their corresponding WT control counterparts were used for the expression analysis of *RP* genes. The heterozygous T-DNA mutant lines were grown in a growth chamber and seeds were harvested. 

### 4.3. Nucleotide Sequence Retrieval of RPS and RPL Genes of Rice and Arabidopsis

The sequences of ribosomal protein large and small subunit genes of rice (*RPL* and *RPS*) were retrieved from RGAP-DB. The sequences were validated using RAP-DB, NCBI and various other databases to ensure that the sequences were gene specific as described by Moin et al. (2016 and 2017) and Saha et al. (2017) [35,36,37]. The *Arabidopsis thaliana RPL* and *RPS* gene sequences were retrieved from Ensemble and the NCBI database and validated using the TAIR database. 

### 4.4. Realtime-qPCR (RT-qPCR) Analyses

Total RNA was extracted from the 7 d-grown plants of *TOR*-OE rice transgenic lines TR-2.24 and TR-15.1 [23], Arabidopsis *TOR*-OE transgenic lines ATR-1.4.27 and ATR-3.7.32 [25] and their corresponding WT plants using Tri-Reagent (Takara Bio, London, UK) following the manufacturer’s protocol. The quality and quantity of extracted RNA were checked on 1.2% agarose gel prepared in TBE (Tris-borate-EDTA) buffer and quantified using a Nano-Drop Spectrophotometer 2000 (Thermo Scientific, Waltham, MA, USA). Total RNA (2 μg) was used to synthesize the first strand cDNA using SMART^TM^ MMLV Reverse Transcriptase (Takara Bio, London, UK). The *Actin1* and *β*-*tubulin* genes were used as endogenous reference genes in RT-qPCR analysis of *RP* genes in the rice transgenic lines, whereas the *Actin2* and α-*tubulin* genes were used as endogenous reference genes for normalization of RT-qPCR analysis of *RP* genes in *TOR*-OE Arabidopsis lines. Specific primers were designed for studying the expression of *RP* genes in Arabidopsis and rice lines using primer3 (v.0.4.0) online tool (Appendix A). The RT-qPCR data were analyzed using three biological and three technical replicates according to the 2^−ΔΔCT^ method [51].

### 4.5. In Silico Prediction of Putative Ser/Thr Phosphorylation Sites in the ribosomal protein Genes of Rice and Arabidopsis

To identify the similarity between the Ser/Thr phosphorylation sites in rice and Arabidopsis RPs, the RPS6, RPL6, RPL18, RPL23, RPL24 and RPS28 protein sequences were retrieved from the databases NCBI, UNIPROT and Ensemble Plants and the retrieved sequences were validated using the TAIR and RAPDB databases. The obtained sequences were then analyzed for the presence of Ser/Thr phosphorylation sites for PKA, PKB, PKC, PKG and RSK protein kinases of the AGC kinase family using NetPhos 3.1 Server (Table 1; http://www.cbs.dtu.dk/services/NetPhos/ accessed on 22 November 2019). The multiple sequence alignment was performed using Clustal Omega (Appendix A; https://www.ebi.ac.uk/Tools/msa/clustalo/ accessed on 17 November 2019) with the selected RPL and RPS proteins to check the similarity between various serine/threonine phosphorylation sites (Table 1; Appendix A).

### 4.6. Genotyping of Arabidopsis Mutants

The *RP* gene mutants *rpl6* (CS16176), *rpl18* (SALK_134424), *rpl23A* (SALK_091329), *rpl24a* (SALK_064513), *rps28A* (SALK_094189), *s6k1* (SALK_113295) and *tor* (SALK_138622) in the Col 0 background were obtained in a heterozygous condition from Arabidopsis Biological Resource Center (Ohio State University, USA). The T-DNA insertions in the mutant lines were confirmed using PCR. These lines were grown by successive inbreeding (the T_1_-generation plants were self-pollinated, the T_2_-seeds were collected, and the process was repeated until collection of T_3_-generation seeds) up to T_3_-generation until they became homozygous insertion lines. As these lines are reported to be embryo lethal when they achieve homozygosity [2,32,33], this shows that the functional nature of the gene amongst their paralogs is essential for embryo development. We, therefore, used the mature seeds from the heterozygous mutant lines in T_2_ generation (their homozygosity was confirmed through genotyping) to check the mRNA levels of the mutated genes in these lines and for protein extraction to detect the S6K1 phosphorylation. The transcript accumulation of genes in the SALK lines was analyzed using RT-qPCR. The mature seeds obtained in T_3_ generation of each SALK line were used to extract total RNA [52]. The cDNAs were synthesized and used to analyze the transcript levels of genes in the homozygous *s6k1*−/−, *tor*−/−*, rpl6*−/−, *rpl18*−/−, *rpl23*−/−, *rpl24*−/− and *rps28*−/− (−/− represents that both alleles were absent in the homozygous mutant lines) SALK lines. The *Actin2* and α-*tubulin* genes were used as endogenous reference genes for normalization and WT was used as the control. The primers used in genotyping and RT-qPCR are detailed in the Appendix A.

### 4.7. Western Blot Analysis

Total protein was extracted from the mature seeds of Arabidopsis WT plants and homozygous T-DNA insertion mutants of Arabidopsis RP genes: *rpl6* (Emb2394, CS16176), *rpl18* (SALK_134424C), *rpl23A* (SALK_091329), *rpl24a* (SALK_064513), *rps28A* (SALK_094189), *Ats6k1* (SALK_113295) and *tor* (SALK_138622), in T_3_-generation using a standard phenol extraction method. As reported earlier, the homozygous *tor−*/*−* and *rpl23aA−*/*−* knockout mutants had defective embryo development [2,32]. To avoid phenotypic aberrations in the homozygous mutants, we used seeds harvested from T_2_ plants for S6K1 phosphorylation assay. The protein sample from the WT-Col 0 was used as a positive control, whereas protein samples from *tor* and *s6k1* mutants were used as negative controls in the phosphorylation assay. The protein precipitate was dissolved in rehydration buffer (7 M Urea, 2 M Thiourea, 4% CHAPS and 30 mM DTT) and quantified using the Bradford method with BSA standards. About 30 μg of total protein was loaded in SDS-PAGE and Western blot analysis. The phosphorylation site of human and Arabidopsis p70kDa-S6K1 proteins is homologous and conserved for Thr-389 and Thr-449 residues [28]. We further performed S6K1 phosphorylation in homozygous T-DNA insertion mutants of the genes encoding for cytosolic ribosomal proteins *rpl6*−/−, *rpl18*−/−, *rpl23A*−/−, *rpl24a*−/−, *rps28A*−/−, *s6k1*−/− and *tor*−/− to determine the cross-link between TOR and RPs. The anti-human S6K1 antibody raised in mice (anti phospho70kDa-S6K1-Thr(P)-389) (Cell Signaling Technologies, Danvers, MA, USA; cat# 9206) was used for detecting S6K1 phosphorylation in the mutants, and WT control, the anti-70kDa S6K1 (CST, Danvers, MA, USA; cat# 9202) and anti-GAPDH (Santa Cruz, Dallas, Texas, USA; FL-335#SC25778) were used as loading controls. The membrane was incubated with secondary HRP-conjugated antibodies and signals were detected using a chemi-luminescence method (ChemiDoc XRS, Bio-Rad, USA).

### 4.8. Identification of Interaction between TOR and Ribosomal Proteins

A protein–protein interactions (PPI) between the TOR protein and RPs were identified through the bioinformatic approach using STRING v11 [53]. The STRING v11 (https://string-db.org/ accessed on 24 July 2021) identifies the interaction based on the evidence from Arabidopsis homologs in other species. The search tool of STRING v11 was used for the retrieval of interacting genes/proteins associated with TOR and ribosomal proteins and to construct the PPI network. The *TOR* and *RP* genes used in the study were listed as inputs in the STRING dB and were searched for their neighbor interactors at a high level of confidence (sources: experiments, databases; score ≥ 0.90).

### 4.9. Bioinformatic Analysis

The results from expression analyses of RP genes were depicted as heatmaps using the MORPHEUS program (https://software.broadinstitute.org/morpheus/ accessed on 24 November 2020). Venn diagrams were depicted using VENNY 2.1 (https://bioinfogp.cnb.csic.es/tools/venny/index.html accessed on 17 November 2020). Quantification of phosphorylation of S6K1 protein in the Western blot analysis was performed using Image J software (https://imagej.nih.gov/ij/index.html accessed on 17 November 2020). The putative Ser/Thr phosphopeptide sequences were identified using NetPhos 3.1 Server (http://www.cbs.dtu.dk/services/NetPhos/ accessed on 22 November 2019) and STRING v11 (https://string-db.org/ accessed on 24 July 2021) was used to predict protein–protein interactions.

### 4.10. Statistical Analysis

Statistical analysis was performed with the mean values using one-way ANOVA in the SigmaPlotv11 software. Mean values are represented with standard error (±SE) with the significance level *p* < 0.001 represented as asterisks ‘**’ and *p* < 0.05 represented as an asterisk ‘*’.

## Figures and Tables

**Figure 1 plants-12-00176-f001:**
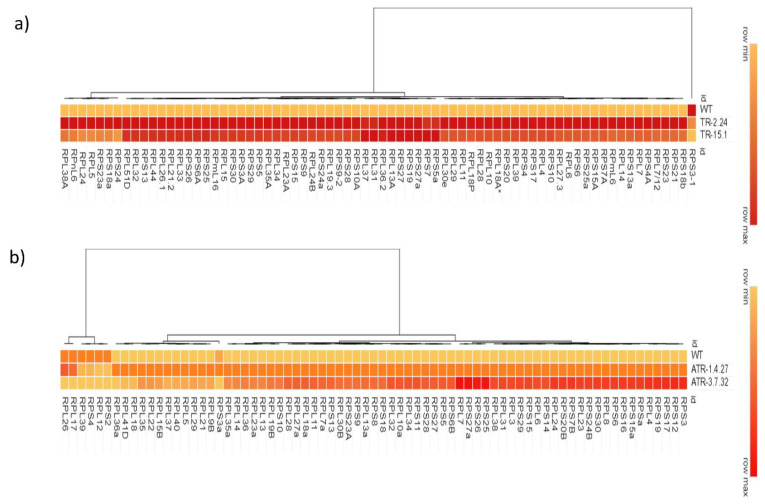
Heatmap representation of expression of *RP* genes in the *TOR*-OE lines. Heatmap represents hierarchical clustering of the expression of *RPL* and *RPS* genes in *TOR*-overexpressing lines of rice and Arabidopsis. The RT-qPCR is used to determine the expression levels of (**a**) *RPL* and *RPS* genes in the *TOR*-OE lines TR-2.24 and TR-15.1 of rice and (**b**) in the lines ATR-1.4.27 and ATR-3.7.32 of Arabidopsis. Red and orange colors indicate increased *RP* gene expression, and the yellow color indicates decreased *RP* gene expression. The fold change was normalized using the 2^−∆∆CT^ method relative to their corresponding WT control.

**Figure 2 plants-12-00176-f002:**
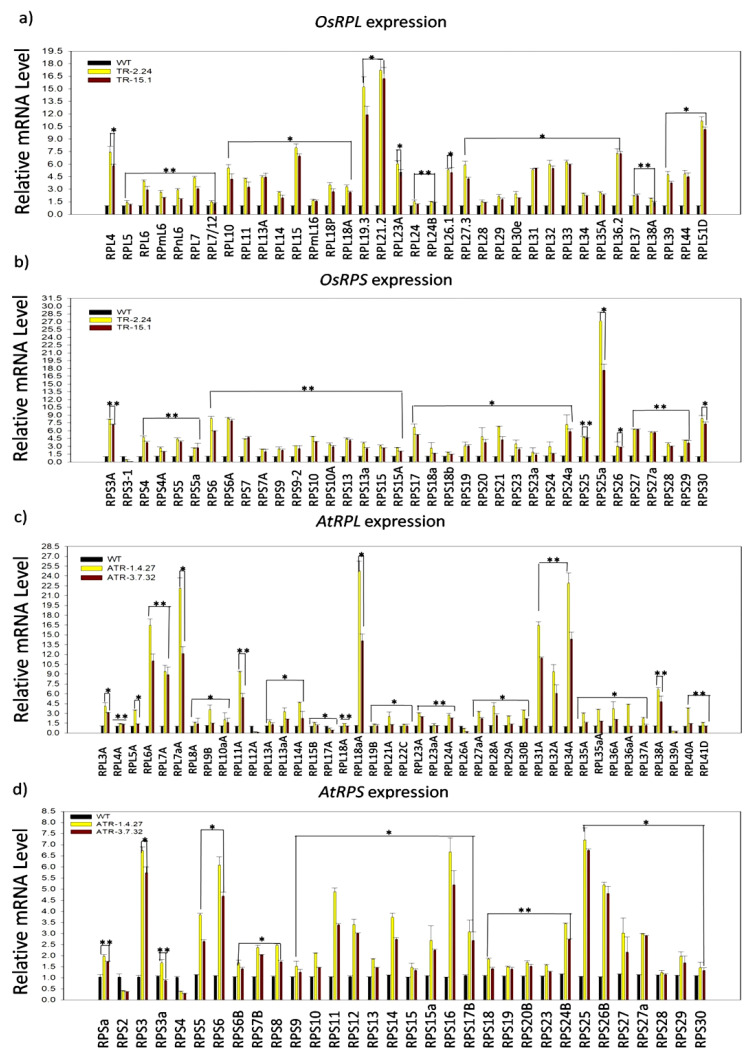
Transcriptional regulation of ribosomal protein large subunit (*RPL*) and small subunit (*RPS*) genes in the *TOR*-OE lines of rice and Arabidopsis. The 7 DAG plants of two high *AtTOR*-expressing rice transgenic lines TR-2.24 and TR-15.1 were used to analyze the expression of *RPL* and *RPS* genes. The WT plants were used as controls. (**a**) Expression analysis of rice *RPL* genes in two high *AtTOR* overexpression lines of rice, TR-2.24 and TR-15.1. The *RPL4*, *RPL14*, *RPL18A*, *RPL19.3*, *RPL36.2* and *RPL51* genes were highly upregulated by 20-fold in both of the transgenic lines. (**b**) Expression analysis of rice *RPS* genes in two transgenic lines. The significant upregulation of *RPS* gene transcripts in two transgenic lines was observed, where the *RPS3A*, *RPS6*, *RPS6A*, *RPS25A* and *RPS30* genes were highly upregulated by more than 7-fold in the transgenic plants. (**c**,**d**) The expression of *RPL* and *RPS* genes was also analyzed in two *TOR*-OE transgenic lines, ATR-1.4.27 and ATR-3.7.32, of Arabidopsis. The fold change was normalized using the ΔΔCT method relative to the WT plants. Rice *Actin* (*Act1*) and Arabidopsis *Actin* (*Act2*) were used as internal controls. Three biological and three technical replicates were included in this study. Vertical bars indicate the mean ± SE of three independent experiments and ANOVA analysis indicated the statistically significant differences, represented by asterisks (*) *p* < 0.05 and (**) *p* < 0.001.

**Figure 3 plants-12-00176-f003:**
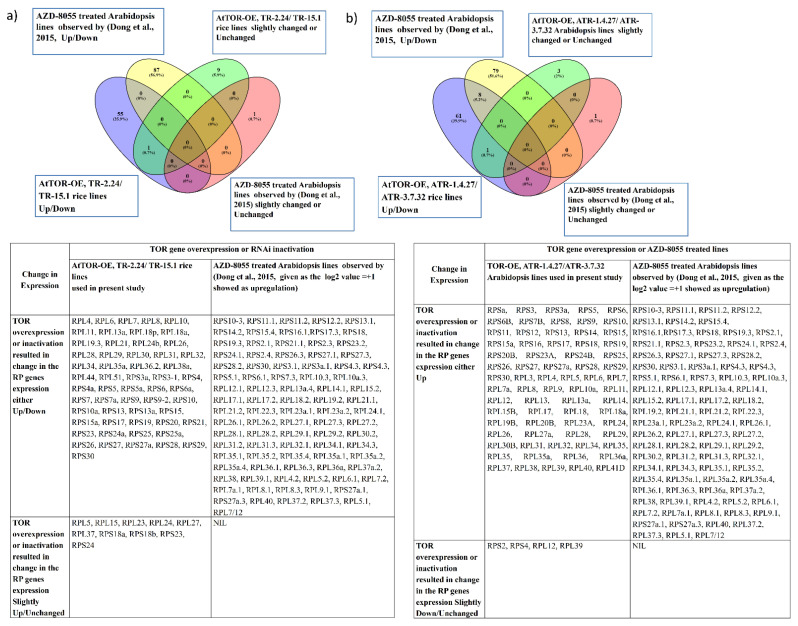
Comparison of the overlapping expression patterns of *RPL* and *RPS* genes in *TOR*-OE lines of rice and Arabidopsis with the AZD-8055-treated Arabidopsis lines. The *RP* genes exhibiting a transcript level of ≥1-fold on log_2_ scale were considered as significantly upregulated and the transcript level below 1-fold was considered downregulated or unchanged in expression. Venn diagrams are used to show the overlaps between the *RP* gene expression (**a**) in the AZD-8055-treated lines of Arabidopsis (Dong et al., 2015 [27]) and *TOR*-OE lines of rice (TR-2.24 and TR-15.1) and (**b**) in the AZD-8055-treated lines of Arabidopsis and *TOR*-OE Arabidopsis lines, ATR-1.4.27 and ATR-3.7.32.

**Figure 4 plants-12-00176-f004:**
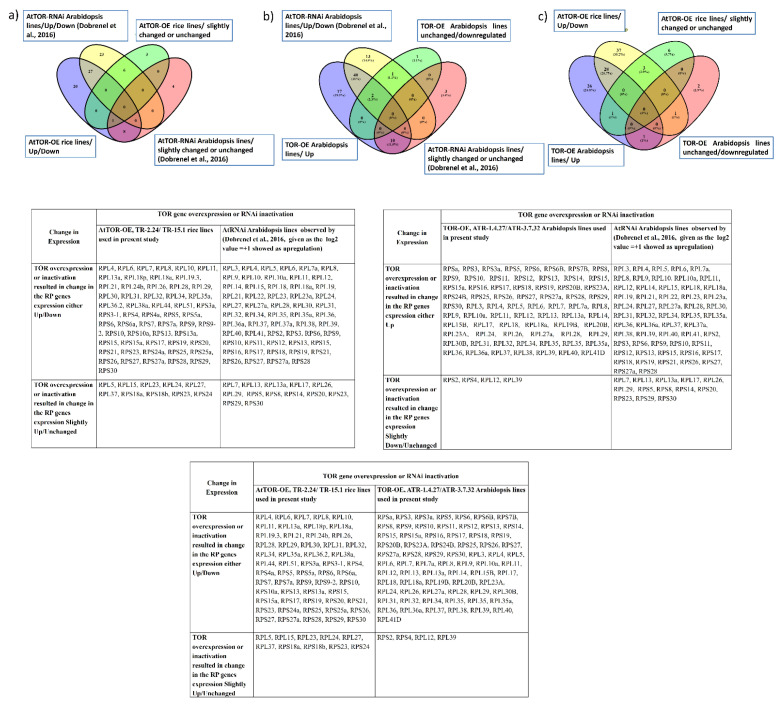
Comparison of the overlapping expression patterns of *RPL* and *RPS* genes in *TOR*-OE lines of rice and Arabidopsis with the *AtTOR*-RNAi Arabidopsis lines. The *RP* genes exhibiting a transcript level of ≥1-fold on log_2_ scale were considered as significantly upregulated and the transcript level below 1-fold was considered downregulated or unchanged in expression. Venn diagrams are used to show the overlaps between the *RP* gene expression (**a**) in the *TOR*-RNAi lines of Arabidopsis (Dobrenel et al., 2016 [19]) and *TOR*-OE lines of rice, TR-2.24 and TR-15.1, (**b**) in the *TOR*-RNAi lines of Arabidopsis and *TOR*-OE Arabidopsis lines, ATR-1.4.27 and ATR-3.7.32, and (**c**) in the *TOR*-OE lines of rice, TR-2.24 and TR-15.1, and ATR-1.4.27 and ATR-3.7.32 Arabidopsis transgenic lines.

**Figure 5 plants-12-00176-f005:**
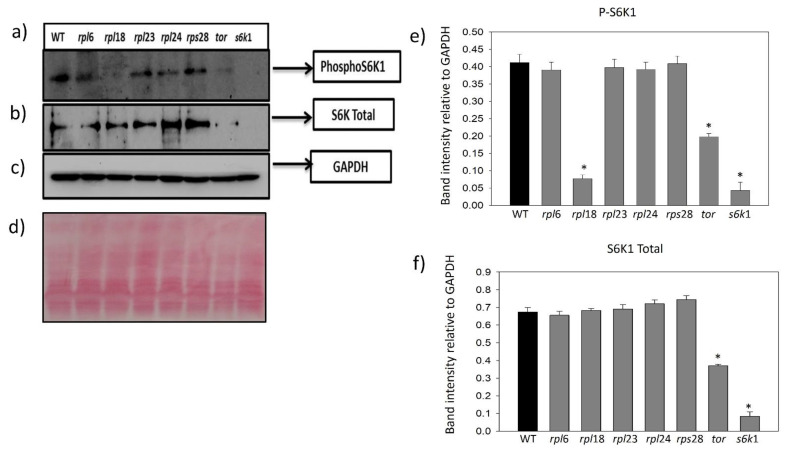
S6K1 phosphorylation assay in Arabidopsis T- DNA insertional mutants. Phosphorylation of p70kDa S6K1 in Thr389 residue was detected in Arabidopsis T-DNA insertional mutants of *tor* and *s6k*1 protein along with mutants of ribosomal proteins *rpl6*, *rpl18, rpl23a, rpl24* and *rps28,* and total protein isolated from WT (Col 0) Arabidopsis was taken as the control. (**a**) Phospho S6K1 detection in all of the mutants; (**b**) Western blot analysis of total S6K protein; (**c**) GAPDH protein used as loading control; (**d**) ponceau stain of the blot (**e**) the relative band intensity of S6K1 phosphorylation and (**f**) total S6K1 to band intensity of GAPDH was analyzed in the Arabidopsis T-DNA insertion mutants using Image J software. One-way ANOVA analysis was performed using the mean values (n = 3) with ±SE. Significantly reduced S6K1 phosphorylation compared with Col0 (WT, control) at *p* < 0.001 are marked with asterisks (*).

**Figure 6 plants-12-00176-f006:**
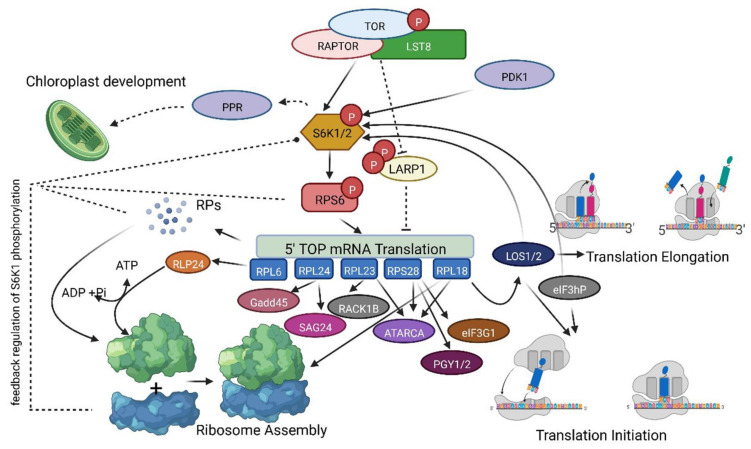
Possible feedback regulation of S6K1 phosphorylation via the TOR pathway. An illustration of TOR Complex 1-mediated regulation of S6K1 phosphorylation and translational initiation by further phosphorylation and activation of RPS6 protein and other RPL and RPS proteins. Dashed lines represent signaling pathways or intermediates that are not fully revealed. The observations from the predicted PPI networks and the inhibition of *RP* genes suggest that the S6K1 phosphorylation is differentially regulated. Possibly, the TOR and RPs are interlinked for the regulation of S6K1 phosphorylation, where RPs also have an independent role in differentially regulating the S6K phosphorylation and modulating protein translation in the plant cell. The figure represents a model for the regulation of S6K1 phosphorylation by loss of RPs in plants, which is possibly mediated via the association of RPs with the S6K1 protein or the other regulatory proteins in the TOR pathway. The illustration was created using www.biorender.com (accessed on 21 December 2022) and exported using a free trial subscription.

**Table 1 plants-12-00176-t001:** Identification of Ser/Thr phosphorylation sites in RPs mediated by the AGC kinase family (e.g., PKC, PKG, RSK, PKA, PKB).

S. No.	Locus Name	Protein Accession	Protein Name	Species Name	Position and Peptide Sequence of Threonine (Thr)-Specific Phosphorylation Sites for AGC Kinases (PKC, PKG, RSK, PKA, PKB)	Position and Peptide Sequence of Serine (Ser)-Specific Phosphorylation Sites for AGC Kinases (PKC, PKG, RSK, PKA, PKB)	Total No. of Ser/Thr Phosphorylation Sites for AGC Kinases (PKC, PKG, RSK, PKA, PKB)	Amino Acid
1	AT4G31700	O48549	AtRPS6A	*Arabidopsis thaliana*	9/VANPTTGCQ 10/ANPTTGCQK 69/QGVLTPGRV 81/LHRGTPCFR 91/HGRRTGERR 127/LPGLTDTEK 129/GLTDTEKPR 157/DDVRTYVNT 161/TYVNTYRRK 167/RRKFTNKKG 185/QRLVTPLTL 188/VTPLTLQRK 249/KPSVTA---	33/DKRISQEVS 37/SQEVSGDAL 98/RRRKSVRGC 105/GCIVSPDLS 109/SPDLSVLNL 141/PKRASKIRK 175/GKEVSKAPK 208/AKANSDAAD 219/KLLASRLKE 229/RDRRSESLA 231/RRSESLAKK 237/AKKRSRLSS 240/RSRLSSAAA 241/SRLSSAAAK 247/AAKPSVTA-	28	250
2	AT5G10360	P51430	AtRPS6B	*Arabidopsis thaliana*	9/VANPTTGCQ 10/ANPTTGCQK 69/QGVLTPGRV 81/LHRGTPCFR 91/HGRRTGERR 127/LPGLTDTEK 129/GLTDTEKPR 161/KYVNTYRRT 165/TYRRTFTNK 167/RRTFTNKKG 185 /QRLVTPLTL 188/VTPLTLQRK	33/DKRLSQEVS 37/SQEVSGDAL 98/RRRKSVRGC 105/GCIVSPDLS 109/SPDLSVLNL 121/KKGVSDLPG 141/PKRASKIRK 175/GKKVSKAPK 208/AKANSDAAD 219/KLLASRLKE 229/RDRRSESLA 231/RRSESLAKK 237/AKKRSRLSS 240/RSRLSSAPA 241/SRLSSAPAK	27	249
3	LOC_Os07g0622100	Q8LH97	OsRPS6	*Oryza sativa* ssp. *japonica*	9/IANPTTGCQ 10/ANPTTGCQK 69/QGVLTAGRV 81/LHRGTPCFR 127/LPGLTDTEK 129/GLTDTEKPR 161/KYVNTYRRT 165/TYRRTFTTK 167/RRTFTTKNG 168/RTFTTKNGK 185/QRLVTPLTL 188/VTPLTLQRK 248/KAAATTA-- 249/AAATTA---	33/DKRISQEVS 37/SQEVSGDAL 98/RRRKSVRGC 105/GCIVSQDLS 109/SQDLSVINL 141/PKRASKIRK 150/LFNLSKDDD 175/GKKVSKAPK 208/AKKKSEAAE 229/RERRSESLA 231/RRSESLAKR 237/AKRRSKLSS 240/RSKLSSAAK 241/SKLSSAAKA	28	250
4	LOC_Os03g27260	Q75LR5	OsRPS6	*Oryza sativa* ssp. *japonica*	9/IANPTTGCQ 10/ANPTTGCQK 69/QGVLTSGRV 81/LHRGTPCFR 127/LPGLTDTEK 129/GLTDTEKPR 161/KYVNTYRRT 165/TYRRTFTTK 167/RRTFTTKNG 168/RTFTTKNGK 185/QRLVTPLTL 188/VTPLTLQRK 243/LSAATTA-- 244/SAATTA---	33/DKRISQEVS 37/SQEVSGDAL 70/GVLTSGRVR 98/RRRKSVRGC 105/GCIVSQDLS 109/SQDLSVINL 141/PKRASKIRK 175/GKKVSKAPK 208/AKKKSEAAE 229/RERRSESLA 231/RRSESLAKR 237/AKRRSKLSA 240/RSKLSAATT	27	245
5	AT1G18540	Q9FZ76	AtRPL6A	*Arabidopsis thaliana*	7/AAKRTPKVN 83/KPKPTKLKA 90/KASITPGTV 93/ITPGTVLII 121/LLLVTGPFK 142/YVIGTSTKI 144/IGTSTKIDI 153/SGVNTEKFD 172/KKKKTEGEF 197/EDQKTVDAA	24/VGKYSRSQM 26/KYSRSQMYH 88/KLKASITPG 114/LKQLSSGLL 115/KQLSSGLLL 143/VIGTSTKID 149/KIDISGVNT 205/ALIKSIEAV 221/GARFSLSQG 223/RFSLSQGMK	20	233
6	AT1G74060	Q9C9C6	AtRPL6B	*Arabidopsis thaliana*	12/AKQRTAKVN 84/PNRRTAKPA 95/RASITPGTV 98/ITPGTVLII 126/LLLVTGPF 147/YVIGTSTKV 149/IGTSTKVDI 157/ISGVTLDKF 177/KKKKTEGEF 219/PELKTYLGA	1/----SPQCC 29/VGKYSRSQM 31/KYSRSQMYH 57/HDAKSKVDA 93/KLRASITPG 120/KQLASGLLL 148/VIGTSTKVD 154/KVDISGVTL 226/GARFSLKQG	19	233
7	LOC_Os04g0473400	Q7XR19	OsRPL6	*Oryza sativa* ssp. *japonica*	4/-MAPTSKLS 19/SRSHTYHRR 66/RQPSTRKPN 72/KPNPTKLRS 79/RSSITPGTV 82/ITPGTVLIL 110/LLLVTGPFK 131/YVIATSTKV 133/IATSTKVDI 161/KAKKTEGEL 168/ELFETEKEA 173/EKEATKNLP 203/PDLKTYLGA	5/MAPTSKLSQ 8/TSKLSQGIK 15/IKKASRSHT 17/KASRSHTYH 65/PRQPSTRKP 76/TKLRSSITP 77/KLRSSITPG 104/KQLKSGLLL 132/VIATSTKVD 138/KVDISGVNV 151/DKYFSRDKK 210/GARFSLRDG	25	222
8	At5g27840	A0A178UKW4	AtRPL18e/L15P	*Arabidopsis thaliana*	14/KSKKTKRTA 17/KTKRTAPKS 72/VEFMTGKDD 84/VLVGTITDD 86/VGTITDDLR 100/AMKVTALRF 105/ALRFTERAR 121/GECLTFDQL 135/LGQNTVLLR 162/PHSNTKPYV	11/AGGKSKKTK 21/TAPKSDDVY 40/LVRRSNSNF 42/RRSNSNFNA 55/RLFMSKVNK 63/PLSLSRL 65/PLSLSRLVE 144/GPKNSREAV 160/GVPHSNTKP 182/GKRKSRGFK	20	127
9	At5g27850	A0A1P8BGQ0	AtRPL18e/L15	*Arabidopsis thaliana*	19/VEFMTGKDD 31/VLVGTITDD 33/VGTITDDLR 47/AMKVTALRF 52/ALRFTERAR 68/GECLTFDQL 82/LGQNTVLLR 109/PHSNTKPYV	2/---MSKVNK 10/ KAPLSLSRL 12/PLSLSRLVE 91/GPKNSREAV 107/GVPHSNTKP 129/GKRKSRGFK	15	134
10	At5g27850	Q940B0	AtRPL18C	*Arabidopsis thaliana*	14/KSKKTKRTA 17/KTKRTAPKS 72/VEFMTGKDD 84/VLVGTITDD 86/VGTITDDLR 100/AMKVTALRF 105/ALRFTERAR 121/GECLTFDQL 135/LGQNTVLLR 162/PHSNTKPYV	11/AGGKSKKTK 21/TAPKSDDVY 40/LVRRSNSNF 42/RRSNSNFNA 55/RLFMSKVNK 63/KAPLSLSRL 65/PLSLSRLVE 144/GPKNSREAV 160/GVPHSNTKP 182/GKRKSRGFK	20	187
11	LOC_Os01g54870	Q943F3	OsRPL18A	*Oryza sativa* ssp. *japonica*	18/RGLPTPTDE 20/LPTPTDEHP 34/KLWATNEVR 72/EKNPTTIKN 73/KNPTTIKNY 87/YQSRTGYHN 99/EYRDTTLNG 100/YRDTTLNGA 110/EQMYTEMAS 128/QIIKTATVH 130/IKTATVHFK 141/KRDNTKQFH 162/VRPPTRKLK 167/RKLKTTFKA 168/KLKTTFKAS	41/VRAKSKFWY 56/KVKKSNGQI 85/LRYQSRTGY 114/TEMASRHRV 147/QFHKSDIKF 172/TFKASRPNL	21	178
12	At2g39460	Q8LD46	AtRPL23aA	*Arabidopsis thaliana*	8/AKVDTTKKA 9/KVDTTKKAD 40/KKIRTKVTF 43/RTKVTFHRP 49/HRPKTLTKP 51/PKTLTKPRT 55/TKPRTGKYP 64/KISATPRNK 80/KYPLTTESA 81/YPLTTESAM 93/EDNNTLVFI 119/YDIQTKKVN 124/KKVNTLIRP 131/RPDGTKKAY 139/YVRLTPDYD	2/---MSPAKV 27/KAVKSGQAF 62/YPKISATPR 83/LTTESAMKK	19	154
13	At3g55280	Q9M3C3	AtRPL23aB	*Arabidopsis thaliana*	8/AKVDTTKKA 9/KVDTTKKAD 40/KKIRTKVTF 43/RTKVTFHRP 49/HRPKTLTKP 51/PKTLTKPRT 55/TKPRTGKYP 64/KISATPRNK 80/KYPLTTESA 81/YPLTTESAM 93/EDNNTLVFI 119/YDIQTKKVN 124/KKVNTLIRP 131/RPDGTKKAY 139/YVRLTPDYD	2/---MSPAKV 27/KAVKSGQAF 62/YPKISATPR 83/LTTESAMKK	19	154
14	LOC_Os03g04590	Q10S10	OsRPL23A	*Oryza sativa* ssp. *japonica*	25/PVAATVNCA 32/CADNTGAKN 64/MVMATVKKG 118/GSAITGPIG	2/---MSKRGR 9/GRGGSAGNK 17/KFRMSLGLP 41/LYIISVKGI 54/NRLPSACVG 115/EMKGSAITG 134/PRIASAANA	11	140
15	LOC_Os10g32920	Q9AV77	OsRPL23B	*Oryza sativa* ssp. *japonica*	25/PVAATVNCA 32/CADNTGAKN 64/MVMATVKKG 118/GSAITGPIG	2/---MSKRGR 9/GRGGSAGNK 17/KFRMSLGLP 41/LYIISVKGI 54/NRLPSACVG 115/EMKGSAITG 134/PRIASAANA	11	140
16	At2g36620	Q42347	AtRPL24A	*Arabidopsis thaliana*	5/MVLKTELCR 54/KLCWTAMYR 77/RRRATKKPY 89/IVGATLEVI 122/RIKKTKDEK	11/LCRFSGQKI 26/RFIRSDSQV 28/IRSDSQVFL 36/LFLNSKCKR 49/KLKPSKLCW 82/KKPYSRSIV 84/PYSRSIVGA 135/VEYASKQQK 140/KQQKSQVKG 149/NIPKSAAPK	15	164
17	At3g53020	P38666	AtRPL24B	*Arabidopsis thaliana*	5/MVLKTELCR 54/KLAWTAMYR 77/RRRATKKPY 89/IVGATLEVI 122/RIKKTKDEK	11/LCRFSGQKI 26/RFIRSDSQV 28/IRSDSQVFL 36/LFLNSKCKR 49/KLKPSKLAW 82/KKPYSRSIV 84/PYSRSIVGA 135/EFASKQQK 151/AAAASKGPK	14	163
18	LOC_Os05g40820/ Os01g0815800	Q5N754-1	OsRPL24A	*Oryza sativa* ssp. *japonica*	5/MVLKTELCR 52/PAKLTWTAM 54/KLTWTAMYR 76/KRRRTTKKP 77/RRRTTKKPY 122/RIKKTKDEK 134/KAEVTKSQK	11/LCRFSGQKI 28/IRADSQVFL 36/LFANSKCKR 82/KKPYSRSIV 84/PYSRSIVGA 89/IVGASLEVI 136/EVTKSQKSQ 139/KSQKSQSKG 141/QKSQSKGAA 149/APRGSKGPK	17	161
19	LOC_Os07g12250	LOC_Os07g12250.1	OsRPL24A	*Oryza sativa* ssp. *japonica*	5/MVLKTELCR 52/PAKLTWTAM 54/KLTWTAMYR 76/KRRRTTKKP 77/RRRTTKKPY 89/IVGATLEVI 122/RIKKTKDEK	11/LCRFSGAKI 28/IRADSQVFL 34/VFLFSNSKC 36/LFSNSKCKR 82/KKPYSRSIV 84/PYSRSIVGA 110/AARESALRE 136/EVAKSQKAS 140/SQKASGKGN	16	161
20	LOC_Os01g33050/	LOC_Os01g33050.2/Q84ZF9	OsRPL24B	*Oryza sativa* ssp. *japonica*	12/FCSSTIYPG 52/KVKWTKAYR 64/GKDMTQDST 68/TQDSTFEFE 86/DRNVTAQTL 89/VTAQTLKAI 97/IPLITKIRH 109/KKHITERQK 117/KQGKTKQRE 139/PKKDTMLST 143/TMLSTQKTK 146/STQKTKVVV 157/SQQQTEENL	10/CWFCSSTIY 11/WFCSSTIYP 34/RFCRSKCHK 67/MTQDSTFEF 142/DTMLSTQKT 153/VVKVSQQQT	19	164
21	LOC_Os07g19190	LOC_Os07g19190.1	RPL24B	*Oryza sativa* ssp. *japonica*	12/FCSSTVYPG 52/KVKWTKAYR 64/GKDMTQDST 68/TQDSTFEFE 86/DRNVTEQTL 89/VTEQTLKAI 97/ISLITKIRH 109/KKHITERQK 117/KQGKTKQRE 139/PKKVTLSTQ 142/VTLSTQKTK 145/STQKTKVVV 156/SQQQTEENL	10/CWFCSSTVY 11/WFCSSTVYP 34/RFCRSKCHK 67/MTQDSTFEF 94/LKAISLITK 141/KVTLSTQKT 152/VVKVSQQQT	20	163
22	At3g10090/At5g03850	Q9SR73	AtRPS28A/B	*Arabidopsis thaliana*	17/VMGRTGSRG 24/RGQVTQVRV 31/RVKFTDSDR 52/GDILTLLES	3/--MDSQIKH 19/GRTGSRGQV 33/KFTDSDRYI 56/TLLESEREA	8	64
23	AT5G64140.1	P34789-1	AtRPS28C	*Arabidopsis thaliana*	17/VMGRTGSRG 24/RGQVTQVRV 31/RVKFTDSDR 52/GDILTLLES	3/--MDSQIKH 19/GRTGSRGQV 33/KFTDSDRYI 56/TLLESEREA	8	64
24	LOC_Os10g0411700	A0A0P0XU17	OsRPS28e	*Oryza sativa* ssp. *japonica*	9/RSMDTQVKL 23/VMGRTGSRG 30/RGQVTQVRV 59/GDILTLLES	6/PEKRSMDTQ 25/GRTGSRGQV 63/TLLESEREA	7	71

The peptide sequences and serine/threonine phosphorylation sites were identified in the Arabidopsis and rice RPL6, RPL18, RPL23, RPL24 and RPS28 protein sequences and similarity were made with the known peptide sequences and serine/threonine phosphorylation sites of RPS6 protein using NetPhos 3.1 Server.

## Data Availability

Data availability is reported in the text. All the data in the study is available in the main text and in the Appendix A.

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
