# Peer review of "Involvement of Target of Rapamycin (TOR) Signaling in the Regulation of Crosstalk between Ribosomal Protein Small Subunit 6 Kinase-1 (RPS6K-1) and Ribosomal Proteins"

_plants, 2023, doi:10.3390/plants12010176_

Round 1
Reviewer 1 Report
In the manuscript "Involvement of Target of Rapamycin (TOR) signaling in the regulation of crosstalk between Ribosomal Protein Small Subunit 6 Kinase-1 (RPS6K-1) and ribosomal proteins", authors analyze the expression levels of RP in Arabidopsis and rice TOR OE transgenic lines. In order to dissect TOR regulation on ribosome biogenesis they analyzed mRNA levels of different RP proteins in WTs and OEs lines. They found a strong expression of the different RP proteins in OEs lines indicating that the expression levels of these genes are under TOR signaling control. To further characterize that, they also analyze the degree of expression (RPL and RPS) overlap between different data sets, including TOR inhibitors treatment or iRNA lines. They also found an interesting feedback loop regulation of RP proteins expression mediated by S6K.
Overall, the control of ribosome biogenesis is not new to TOR community (Powers and Walter, 1999) however it is true that this study presents a deeper analysis of the different RP proteins and shows pieces of evidence on how this regulation takes place in green organisms. However, it needs some writing improvement as it is presented in a very extended version that needs to be shortened and focus on the author´s main objective. In this sense, the abstract and introduction need to focus on what is the real discovery of the data presented in this manuscript.
The overall quality of the figures needs to be improved, especially those showing overlap among conditions. In this aspect, I strongly recommend moving figures 5, 6, and 7 to the supplemental material.
Figure 8 panels E and F need some readjustment of the labeling as it gets very confusing at the final bars.
Discussion section needs further analysis and comparison with the data obtained in this work. One of the important results is the downregulation of TOR activity in Atrpl18 line as according to figure 8 seems to be very specific to this mutation.
I also recommend updating some references.
Reviewer 2 Report
The manuscript describes the role of Target of Rapamycin (TOR) signaling in transcriptional and translational regulation of Ribosomal proteins (RPs) in rice and Arabidopsis. The paper indicated a correlation between phosphorylation of S6K1 and ribosomal protein function and the regulation of ribosomal proteins and TOR pathway are interlinked via S6K1 phosphorylation. I have some major concerns with this paper:
1. Figure 8 lacks statistical difference analysis. Figure 8a and 8e showed that rpl18 mutant significantly reduced the phosphorylation level of S6K1 protein, while in other ribosomal protein mutants did not change significantly. How to explain this? Is it an experimental problem or does rpl18 have an important effect on the phosphorylation of S6K1 protein? Please confirm it further.
2. RPL18 is not covered in Figure 9. The information needs to be addressed in the Discussion to explain whether RPL18 has unique function.
3. The Figures quality of the manuscript needs to be improved, especially in Figure 3 and Figure 4 (The figures have more text and less font, which makes it unreadable).
4. The full text format needs to be checked throughout, for example strongly reduced write in italics in line 384, and the genes should write in italics in Figure 7.
5. Significance analysis and bioinformatics analysis contents are missing in Material and Methods
Round 2
Reviewer 2 Report
Accept in present form